# The Availability of Essential Antimicrobials in Public and Private Sector Facilities: A Cross-Sectional Survey in a District of North India

**DOI:** 10.3390/antibiotics13020131

**Published:** 2024-01-29

**Authors:** Niti Mittal, Rakesh Mittal, Sukhbir Singh, Sushila Godara

**Affiliations:** 1Department of Pharmacology, Pt. B D Sharma Postgraduate Institute of Medical Sciences, Rohtak 124001, Haryana, India; rakeshmittal.pgims@uhsr.ac.in; 2Department of Hospital Administration, Pt. B D Sharma Postgraduate Institute of Medical Sciences, Rohtak 124001, Haryana, India; drbrar1980@gmail.com; 3Health Department, Government of Haryana, Panchkula 134109, Haryana, India; drgodarasushila@gmail.com

**Keywords:** essential antimicrobials, essential medicine list, drug procurement, gap analysis, pediatric antibiotic formulations

## Abstract

(1) Background: There is a need to assess the availability of essential antimicrobials, as the availability of an antimicrobial is a critical element of its rational use. We aimed to assess the availability of antimicrobials listed in the National List of Essential Medicines 2015, India (primary list), and a selected (secondary) list comprised of agents indicated for commonly encountered infectious illnesses in various healthcare settings and to identify the reasons for their non-availability. (2) Methods: A cross-sectional survey of 25 public, private, and other sector pharmacies was carried out in Rohtak, a district of the North Indian state of Haryana, from April to June 2022. (3) Results: Most of the antimicrobials surveyed were optimally available in various sector pharmacies with the exception of benzathine benzylpenicillin, benzylpenicillin, cloxacillin, cefazolin, cefuroxime, cefadroxil, amphotericin B, and antimalarials. The most frequent reasons for limited availability were low demand, no prescriptions, and the non-listing of drugs in the state’s essential medicine list. (4) Conclusions: Enough evidence needs to be generated with respect to the status of availability of essential antimicrobials from different regions of India as well as other lower-middle-income countries to devise measures for ascertaining better availability of these agents, especially antibiotics at regional, national, and global scales.

## 1. Introduction

The WHO pioneered the concept of essential medicines in 1977 when it introduced the essential medicine list (EML) model [1]. According to the WHO, essential medicines are defined as “*those that satisfy the priority health care needs of the population. They are selected with due regard to public health relevance, evidence on efficacy and safety, and comparative cost-effectiveness*” [2]. The whole idea of essential medicines is to ensure their availability within the context of functioning healthcare systems at all times in sufficient quantities, in appropriate dosage forms, with assured quality and adequate information, and at a price the individual and the community can afford. Various country-specific essential medicine lists have been developed to address the disease burden of respective nations. India produced its first National Essential Drugs List in 1996, which was later revised in 2003, 2011, and 2015 as the National List of Essential Medicines (NLEM) [3]. The latest revision of NLEM was released in September 2022 [4]. In line with NLEM, the majority of the states maintain their state-specific EMLs, which also focus on commonly used medicines at different healthcare levels in their respective areas.

One of the major public health concerns identified in recent times is the lack of regular access to essential medicines, which in turn has significant implications on the prescribing behavior of clinicians, urging them to prescribe less effective, more toxic, and/or more expensive agents leading to increased healthcare costs and poor treatment outcomes [5]. The use of alternative sub-optimal agents due to the non-availability of essential and more effective antimicrobials can result in incomplete eradication of non-susceptible pathogens, which can grow and spread, thus contributing to antimicrobial resistance (AMR), a rapidly emerging global threat [5]. Poor access to essential antimicrobials is also a driver of high mortality, especially in low-middle-income countries [6]. Besides compromised health outcomes, the economic consequences of such limited access are enormous, with an estimated cost of EUR 20–30 million linked to a shortage of just one antimicrobial, as per the WHO report [7]. Hence, there is a need to address the issue of restricted availability of essential antimicrobials, which poses a serious threat to rational antimicrobial use and hinders the attainment of successful antimicrobial stewardship.

Extensive data from middle- to high-income countries report sub-optimal availability of essential antibiotics [8,9,10,11]. In a survey conducted by the European Society of Clinical Microbiology and Infectious Diseases Study Group for Antimicrobial Stewardship (ESGAP), the researchers assessed the availability of a list of antibiotics across 38 countries in Europe, the US, Canada, and Australia. Systemic antibiotics approved by the US Food and Drug Administration (FDA) and/or European Medicine Agency (EMA) and/or in Europe (35 countries), Canada, and Australia were enlisted as antibiotics to be surveyed. It was reported that 22 out of the 33 selected antibiotics were available in less than 20 included countries [8]. Later, in 2015, the updated survey reported an even worse situation, with 25 of 36 selected antibiotics marketed in 20 of 39 countries or less [9]. Quadri et al. reported a shortage of 148 antibacterial drugs in the USA between 2001 and 2013, with 22% of drugs experiencing multiple shortage periods [10]. Another study from Japan reported critical shortages in cefazolin supply with a resultant many-fold increased use of third-generation cephalosporins, specifically cefotaxime and ceftriaxone (both ‘watch’ group antibiotics) [11].

Data on the availability of antimicrobials from low- and middle-income countries (LMICs) are limited and patchy [12,13,14,15]. In a large survey conducted across 36 developing and middle-income countries by Cameron et al., the mean availabilities of amoxicillin 250 mg capsule/tablet and ciprofloxacin 500 mg capsule/tablet were reported as 68.7 and 52% in the public sector and 76 and 82.4% in the private sector, respectively [14]. Knowles et al., in another survey conducted across 20 low- and middle-income countries (LMICs), reported a 48.9% median availability of 27 antibiotics (19 access, 7 watch, 1 unclassified) in all health facilities surveyed [15].

There is a need to analyze the status of the availability of essential medicines in India in order to implement strategies for ensuring better availability of these drugs at local, regional, and national levels. With this background, the present study was planned to generate information on the availability of essential antimicrobials listed in NLEM 2015 in public as well as private sectors in a district of North India and to identify the reasons for their non-availability.

## 2. Results

A total of 25 pharmacies comprising 13 public (7 primary, 5 secondary, and 1 tertiary healthcare), 10 private, and 2 other sector pharmacies were surveyed in the district. Of these, one tertiary, two secondary, one primary, six private, and two other sector pharmacies were located in urban areas, while the rest were in rural and semi-urban areas.

### 2.1. Availability of Antimicrobials Listed in Primary (NLEM 2015) and Secondary (Selected) Lists

#### 2.1.1. Antibiotics

In the public sector, antibiotics with more than 80 percent availability were tablet/capsule formulations of amoxicillin, amoxicillin–clavulanic acid, cefixime, azithromycin, doxycycline, ciprofloxacin, cotrimoxazole, metronidazole, and injection ceftriaxone. Cefazolin, cefuroxime, and clarithromycin were not available in the public sector. Except for cloxacillin and cefazolin, which were available in up to 30 percent of surveyed private retail pharmacies, most of the antibiotics were optimally available in the majority of pharmacies. For most of the antibiotics, availability in other sector pharmacies was parallel to that in the public sector. There was absolute non-availability of benzathine benzylpenicillin and benzylpenicillin across all sectors in the district (Table 1).

#### 2.1.2. Other Antimicrobials

Tablet albendazole and ivermectin were optimally available anthelminthics across all sectors. Among antivirals, acyclovir and oseltamivir were available in the majority of the pharmacies, while clotrimazole pessary and tablet fluconazole, itraconazole, and voriconazole were optimally available antifungals. There was, however, poor availability of amphotericin B, antimalarials, nystatin, other antivirals, and anthelminthics in all surveyed pharmacies (Table 2).

#### 2.1.3. Pediatric Formulations of Antimicrobials

Most of the antibiotic formulations for pediatric use (suspensions and syrup formulations), including amoxicillin, amoxicillin–clavulanic acid, cefadroxil, cefixime, azithromycin, and metronidazole, were available in primary and secondary care public pharmacies. Among such formulations available in tertiary care centers were amoxicillin, amoxicillin–clavulanic acid, azithromycin, and cotrimoxazole. The availability of pediatric antibiotics in private and other sector pharmacies was optimal for most of the agents surveyed. Suspension albendazole was available in more than 70% of facilities surveyed across different sectors. There was poor availability of acyclovir pediatric formulation in the desired tertiary care as well as in private and other sector facilities.

### 2.2. Gap Analysis for the Availability of Antimicrobials in the Public Sector

According to NLEM 2015, the overall availability of anti-protozoals, anthelminthics and antifungals was less than 50 percent across all healthcare levels, while antivirals had sub-optimal overall availability in secondary care facilities (Table 3).

In contrast, there was good overall availability of all antimicrobial classes across all healthcare levels as per the selected list (Table 4).

More than 60 percent of the desired ‘Access’ group of antibiotics was available in primary and secondary healthcare facilities, while approximately 70 percent availability of the ‘Watch’ group was observed in either sector. In tertiary care, there was optimal availability (more than 50 percent) for all the AWaRe groups (Figure 1).

### 2.3. Reasons for Sub-Optimal Availability

For many antimicrobials, the reasons for their non-availability were not specified. Low demand/no prescriptions was the most frequently cited reason in either sector, while lack of supply and non-listing of the drug(s) in the state EML were the other common reasons quoted for their limited availability in the public sector (Figure 2).

## 3. Discussion

Access to essential medicines is a crucial element of universal health coverage. The present survey was conducted to evaluate the availability of essential antimicrobials included in NLEM 2015 and a selected/secondary list for the survey.

For both primary and secondary lists of antimicrobials, overall availability at a particular healthcare level and total overall availability across all levels showed comparable results to the study by Prinja et al. (total overall availability: 59.6% and 47.2% for antibiotics and antifungals, respectively) [12] assessing the availability of medicines listed in state EML and provided under national health programs in addition to NLEM, which may explain the discordance seen for antivirals and anthelminthics/antiprotozoals from the current study. In another survey investigating the availability of a basket of 24 essential and 8 high-end antibiotics across various public and private pharmacies in Delhi, Kotwani et al. reported sub-optimal availability of a few essential antibiotics such as ampicillin suspension, benzathine penicillin, cefixime, etc., in the public sector pharmacies and reasonably good availability of non-essential and high-end antibiotics in both public and private sectors [13].

As per the WHO’s defined common global target, the access group of antibiotics should comprise >60% of overall antibiotic use [16]. Of note, there was optimal availability of most access groups of antibiotics across public and private sectors. Besides these, antibiotics used to treat common infections encountered in primary or secondary care, such as doxycycline, azithromycin, ciprofloxacin, and cefixime, were available in most of the respective pharmacies. These findings are in line with an earlier large survey conducted in 20 LMICs [15].

The issue of poor availability of antibiotics for pediatric use is particularly troublesome owing to the fact that fewer agents are approved by regulatory authorities for use in children as compared to adults, and finding safe, alternative therapies for agents with poor availability is, therefore, quite challenging [17]. In a survey conducted in Odisha, pediatric formulations of antibiotics included in the state EML (amoxicillin–clavulanic acid, amoxicillin, cotrimoxazole azithromycin) were reported to be available in less than 60 percent of public and private facilities [18]. In our survey, the availability of desired pediatric antibiotic preparations was although optimal in most primary and secondary care centers, but it was not so in tertiary care centers. Low demand and fewer prescriptions were reported as the reason for their non-availability, which may partially be explained by the fact that most inpatients in tertiary care referral centers have severe illnesses demanding parenteral administration of drugs. However, whether it is the true demand–supply relationship or vice versa (poor availability affecting the prescribing practices) needs to be further explored. In fact, the intravenous to oral switch of antibiotics is a key strategy defined under antibiotic stewardship, the fruitful implementation of which may be governed by the availability of oral preparations apart from the physicians’ prescribing behaviors. There is limited data regarding the impact of antibiotic shortages on the pediatric population in India. Few researchers from developed nations have reported a higher incidence of acute kidney injury among children associated with piperacillin–tazobactam and vancomycin combination therapy when used as an alternative to anti-pseudomonal beta-lactam antibiotics due to the limited availability of the latter [19,20].

Sub-optimal availability of some antimicrobials such as penicillin, cefazolin, cloxacillin, antimalarials, antivirals, etc., particularly in the public sector, seems alarming at first instance; however, a deeper understanding of the associated factors probably throws some insight into the situation. Commonly cited reasons for the poor availability of most surveyed drugs in public pharmacies were poor supply from district warehouses, which is mainly governed by demand from various pharmacies and, in turn, by the prescribing patterns. For example, the declining trend in reported cases of syphilis in public tertiary care centers in Rohtak [21] may partly account for the very low prescription rate of benzathine penicillin. This is largely conjectural since the aforementioned data were hospital-based and may not be reflective of the true prevalence of syphilis in the community, where most people may be resorting to private practitioners. Cloxacillin is another beta-lactam antibiotic effective against methicillin-sensitive *Staphylococcus aureus,* whose demand and supply are mainly governed by local resistance patterns, a fact that demands a deeper analysis of the link between antibiotic resistance and utilization patterns. Cefazolin and cefuroxime, belonging to the cephalosporin group of antibiotics and recommended for surgical prophylaxis, were not available in public pharmacies with low demand and non-listing of these drugs in Haryana state’s EML being the quoted reasons for non-availability. Non-availability of antimalarial agents in the public sector was explained by the fact that the district of Rohtak (in addition to six other districts, viz. Ambala, Bhiwani, Jind, Kaithal, Karnal, and Kurukshetra in the state of Haryana) was given ‘zero indigenous case’ status in the year 2021 after no case was reported for 1 year.

Extensive use of amphotericin B during the then-recent coronavirus-associated mucormycosis outbreak mainly accounted for the limited availability of this antifungal at the time of the survey conducted in early 2022. However, there is a recovering trend for its availability, and a follow-up survey may provide a better picture regarding this. The lack of supply of anti-leishmanial agents (miltefosine and paromomycin) was ascribed to infrequent reports of leishmaniasis in the district [22] and the drugs being covered under the National Vector Borne Disease Control Programme (NVBDCP) [23]. Similarly, anti-hepatitis drugs (such as interferons, entecavir, ribavirin, sofosbuvir, and tenofovir) are being provided under the National Viral Hepatitis Control Program (NVHCP) [24], explaining their non-availability in public and private pharmacies.

Non-listing of some drugs in the state’s EML was another reason cited for their non-availability; for example, clarithromycin, antivirals except acyclovir, and antiprotozoals except diloxanide are not included in Haryana state EML (2013–2014), [25] and thus had poor availability in the public sector. On the other hand, the availability of some agents used to treat commonly encountered infections, such as nitrofurantoin, ivermectin, oseltamivir, and itraconazole, was reasonably good despite their not being listed in the state’s EML, an observation emphasizing the need to periodically revise the list as per the emerging demands. Ivermectin, cefuroxime, itraconazole, and terbinafine have, however, been included in the latest revision of NLEM released in September 2022 (Appendix A).

A few limitations of this study need to be considered. This study was designed to evaluate the availability of antimicrobials on the respective day of the survey; however, we calculated the overall availability of antimicrobials for the various levels of facilities using standard mathematical formulae. Also, the conduct of a survey across several facilities over a span of 3 months reasonably reflects average availability over time. Secondly, although the survey results provide quantitative data regarding the availability of antimicrobials, other important factors determining access to medicines, such as pitfalls in procurement and/or distribution systems, prescribing practices in different settings, etc., were not taken into account. A concurrent prescription audit might contribute to drawing better conclusions and linking prescribing with local availability, which may be planned in future surveys of such kind. Issues such as the influence of drug pricing and regulation on availability also need to be addressed in future research. Thirdly, availability was assessed as a parameter of the supply of medicines. However, another critical factor influencing availability should be standard treatment guidelines, which is a potentially important research question. Lastly, due to feasibility issues, the survey was conducted in a single district of North India, and hence, the results may not be reflective of the availability patterns in other districts in the state and in the country per se.

Despite the limitations, a few strengths of this study are worth mentioning. The present survey provides a snapshot of the availability of other antimicrobials in addition to antibiotics, which were not taken into account in most of the surveys reported earlier. Both public and private sector pharmacies were covered, while the earlier surveys from India mainly focussed on the public sector. Generating data on the availability of antimicrobials in private retail pharmacies is important to have a more comprehensive picture of the problem, a fact well recognized previously by researchers [12]. Admittedly, a comparative evaluation of the availability of essential versus non-essential/high-end antibiotics as determinants of prescribing practices is an area demanding further investigation. Moreover, there is a pressing need for periodic review of antibiotic resistance patterns and their correlation with antibiotic availability and prescribing behavior.

## 4. Materials and Methods

### 4.1. Study Setting

The present study was conducted in Rohtak, a district in the North Indian state of Haryana, with more than half of the total district population of 1.17 million residing in rural areas [26]. The district has a 3-tier public healthcare delivery system as in the rest of the country: primary (primary health centers (PHCs) and sub-centers), secondary (community health centers (CHCs) and civil and district hospitals), and tertiary levels (medical colleges). Besides the public sector, healthcare is also provided by private clinics, nursing homes, and corporate hospitals. As per the national sample survey data (2017–2018), around two-thirds of the population depends on private sector for treatment, and nearly 70 percent of expenditure on medicines is borne by patients out-of-pocket [27]. In the three North Indian states of Haryana, Punjab, and Chandigarh, public sector caters to the healthcare needs of 59 to 86 percent of the population [28].

### 4.2. Drug Procurement Model in the District

The procurement and distribution of medicines in the public sector in the district Rohtak, Haryana, is by “The Medicine Procurement and Management Policy 2012” [29]. The medicines are procured from funds received from the state government (budget for the purchase of medicines) and the Government of India as part of the National Health Mission (NHM) program. The procurement of medicines is decentralized at district level, whereby the district health societies are authorized to procure medicines and consumables and further distribute these to health facilities. Demand for medicines is put up by the health facilities to the local warehouse or supplier of medicine in the form of an indent. Every health facility maintains a record of demands, stockouts, and utilization statuses of medicines.

### 4.3. Study Design

A cross-sectional survey of medicine outlets across various public and private sectors in the district of Rohtak was carried out to gather information relevant to the study objectives. The types of medicine outlets surveyed included the following: public sector: pharmacies in public primary, secondary, and tertiary care health centers; private sector: private retailers in the community (only licensed pharmacies and drug stores); and other sector: private pharmacies in public hospitals, health facilities run by non-governmental organizations such as charitable organizations, health facilities run by religious organizations, private hospitals, etc.

### 4.4. Survey Facilities Sampling

The pharmacies of the main public hospital(s) in the district, viz. Postgraduate Institute of Medical Sciences (PGIMS) and Civil Hospital (Rohtak), were surveyed. Sampling frame was constructed by compiling the lists of facilities in various sectors in the district as follows: (i) *public sector*: all public health facilities (e.g., primary and community health centers and district/sub-district and civil hospitals) in the district from the website of the Health Department, Haryana (www.haryanahealth.gov.in/, accessed on 10 March 2022); (Appendix A: list of pharmacies in public and private sectors in the district Rohtak) (ii) *private sector*: currently licensed retail pharmacies from the website www.medicineindia.org (accessed on 12 March 2022); and (iii) *other sector*: private pharmacies in public hospitals, health facilities run by non-governmental such as charitable or religious organizations, and private hospitals.

In district Rohtak, the number of pharmacies in public sector is as follows: primary (24 rural and semi-urban PHCs, 4 urban PHCs), secondary (7 CHCs, 1 civil hospital), and tertiary (1 tertiary medical college: Postgraduate Institute of Medical Sciences (PGIMS). At the time of survey, there were a total of 145 private retail pharmacies. The facilities to be surveyed were selected by systematic random sampling. For this, the sampling interval was calculated as the total number of pharmacies (in rural, urban, and semi-urban areas in the district) in the sampling frame (N) divided by the sample of interest (n = 5 for primary and secondary care public pharmacies; 10 for private pharmacies, and 5 for another sector). If the number of public healthcare facilities at any level was fewer than 5, it was planned that the number selected from another level would be increased accordingly. A random number was generated in Excel and multiplied by the derived sampling interval to yield the sample start number. The number so obtained was (if needed) rounded up to the next integer to obtain the serial number of the facility to begin sampling with. For each facility selected, an additional nearest facility was selected to be used as backup when needed, such as in situations where the manager of the facility from the primary sample does not permit data collection even after being shown the required documents.

### 4.5. Data Collection

The data were collected by a team of investigators trained a priori during the period of April to June 2022. Data on the availability of antimicrobials were extracted from stock registers available. Structured and pretested data collection forms were used to record information on the availability/non-availability of listed antimicrobials. A drug was considered available if it was in stock on the day of the survey. Medicines available through vertical health programs, e.g., antitubercular and anti-retroviral drugs, were excluded from the current analysis. The data on the availability of antimicrobials were collected with respect to antimicrobials listed as essential in NLEM 2015 (primary list) (Appendix A) and a selected (secondary) list of antimicrobials, which was prepared by consensus among study investigators after reviewing the agents indicated for commonly encountered infectious illnesses in various healthcare settings (by discussion with medical officers and physicians) (Appendix A). 

### 4.6. Data Analysis

Data were entered into Microsoft Excel and summarized using descriptive statistics. No specific hypothesis was tested. The data were subjected to following analyses.

#### 4.6.1. Availability of Antimicrobials Listed in Primary (NLEM 2015) and Secondary (Selected) Lists

This was expressed as the number (percentage) of facilities (public/private/other) where a particular drug was available on the day of the survey. In the absence of any standard criteria for defining “optimal availability” in the published literature, a drug was considered optimally available for this study if at least 50 percent of the surveyed pharmacies in a particular sector had the drug available on the day of survey.

#### 4.6.2. Gap Analysis for the Availability of Antimicrobials in the Public Sector

For both primary and secondary lists, gap analyses for availability of antimicrobials in public sector were conducted separately.

*Facility-wise availability.* Availability of a particular class of antimicrobials (antibiotics, anthelminthics, antifungals, antivirals, and antiprotozoals) in a facility was calculated as n/N × 100, where n = number of drugs available within that class on the day of the survey, and N = total number of surveyed drugs in that class for the respective sector.

*Overall availability of a particular class of antimicrobials for a particular level of facility.* This was calculated by the formula Σ(n_i_)×100/A×B, where n = number of drugs available within a particular class in a facility, A = total number of drugs in that class that are essential for the respective level of the facility, and B = number of facilities surveyed in that level.

*Total overall availability of a particular class of antimicrobials across all levels of care.* This was obtained using the formula Σ(n_i_) × 100/ΣA_i_ × B_i_, where n = number of drugs available within a particular class in a facility, A = total number of drugs in that class that are essential for the respective level of the facility, and B = number of facilities surveyed in that level.

*Overall percentage availability of all drugs in a facility.* This was calculated as Σ(n_i_) × 100/ΣA_i_, where n = total number of antimicrobials available in a facility, and A = total number of antimicrobials required for the respective level of the facility.

For gap analysis, a drug was considered “available” at a particular facility if at least one dosage was available.

#### 4.6.3. Gap Analysis for Availability of Antibiotics in Public Sector as per WHO Access, Watch, and Reserve (AWaRe) 2021 Classification [30]

From the list of overall antibiotics surveyed in different healthcare levels (primary: 17; secondary: 25; tertiary: 29), gap analysis for access, watch, and reserve groups of antibiotics was conducted separately in a similar manner as described earlier.

#### 4.6.4. Reasons for Sub-Optimal Availability of Antimicrobials

For non-available agents, inquiries regarding the reasons for their non-availability came from the pharmacist/store-in-charge/any other person handling procurement and dispensing of medicines at the facility level through open-ended questions. The reasons for non-availability were noted down and expressed as frequency measures for overall antimicrobials.

### 4.7. Ethical and Administrative Approvals

This study was conducted after obtaining ethical and administrative approvals from the Institutional Ethics Committee of PGIMS, Rohtak, Haryana (vide letter no. BREC/21/50) and Director General Health Services (DGHS) Haryana (vide letter no. 8/65-HE-2021/805), respectively. Before data collection, written informed consent was obtained from the medical officer/chief pharmacist and manager/owner of medicine outlets for public, other, and private sector pharmacies, respectively.

## 5. Conclusions

Increased awareness of the problem is a crucial step toward addressing the issue of poor availability of essential drugs. The present survey conducted in a district of Haryana, a North Indian state, reported optimal availability of most of the surveyed antimicrobials with the exception of a few antibiotics, amphotericin B, and antimalarials. A deeper scrutiny of associated factors (no prescriptions/demand, prevailing resistance patterns, dwindling demand–supply chain for amphotericin B, and ‘zero indigenous case’ status of the district for malaria), however, projects that the situation is not as alarming as it seems at first instance. However, enough evidence needs to be generated in this regard from different regions of India as well as other LMICs; underlying reasons for poor availability should be identified, which may provide guidance in devising measures for ascertaining better availability of antimicrobial agents, especially antibiotics at regional, national, and global scales. For drugs like antibiotics, which may not be profitable but are life-saving, strategies to incentivize the manufacturers need to be implemented to ensure their consistent production and supply. There must be political commitments to strengthen the availability of essential medicines in a long-term and sustainable manner in the public sector, which may help reduce private healthcare expenditures. Innovation, access to existing and time-tested drugs, and judicious usage should go hand in hand to facilitate the rational use of drugs and achieve better clinical and economic outcomes.

## Figures and Tables

**Figure 1 antibiotics-13-00131-f001:**
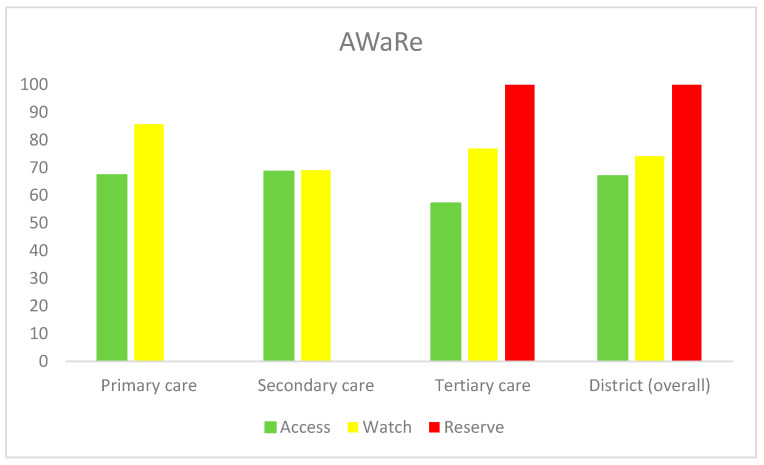
Gap analysis for availability of antibiotics in public sector according to WHO AWaRe classification. Data are represented as percentages. The total number of antibiotics surveyed for different healthcare levels are as follows: primary care (17: 14 access, 3 watch); secondary care (25: 14 access, 11 watch), and tertiary care (29: 14 access, 13 watch, 2 reserve).

**Figure 2 antibiotics-13-00131-f002:**
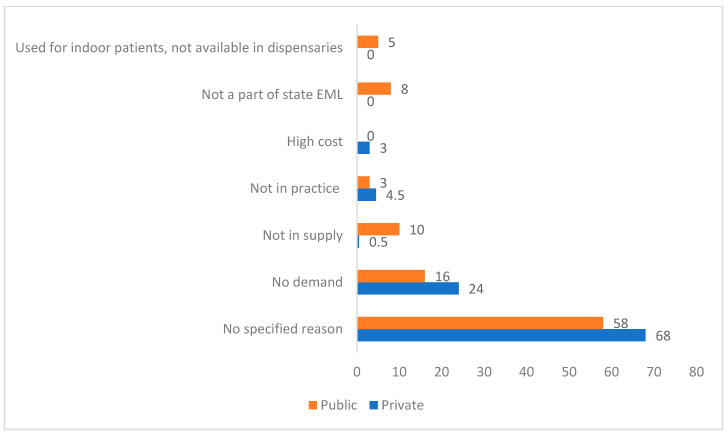
Reasons for non-availability of antimicrobials in public and private sectors.

**Table 1 antibiotics-13-00131-t001:** Availability of antibiotics included in National List of Essential Medicines 2015 and the selected (secondary) list for survey in medicine outlets of different sectors in the district.

S. No.	Name of Antibiotic	WHO AWaRe (Access, Watch, Reserve) Category	Formulation	Strength	Healthcare Facility Level	Data on Availability			
						Public Sector	Private Sector	Other Sector	Total
						Primary care (N = 7); Secondary (N = 5); Tertiary (N = 1)	Private Retailers (N = 10)	(AMRIT * Pharmacy; N = 2)	
Antibiotics included in both NLEM 2015 and the selected (secondary) list for survey									
1.	Amoxicillin	Access	Capsule	250 mg, 500 mg	P, S, T	7 (100)/5 (100)/A	9 (90)	2 (100)	24/25 (96)
			Suspension	125 mg/5 mL	P, S, T	7 (100)/5 (100)/A	9 (90)	1 (50)	23/25 (92)
2.	Amoxicillin + clavulanic acid	Access	Tablet	500/125 mg	P, S, T	5 (71.4)/3 (60)/A	10 (100)	2 (100)	21/25 (84)
			Injection	0.6 g, 1.2 g	S, T	-/3 (60)/A	8 (80)	0	12/18 (66.7)
			Suspension	228.5 mg/5 mL	P, S, T	3 (42.8)/4 (80)/A	9 (90)	2 (100)	19/25 (76)
3.	Cefazolin ^#^	Access	Injection	500 mg, 1 g	P, S, T	0/0/NA	3 (30)	0	3/18 (16.7)
4.	Cefixime	Watch	Tablet	100, 200, 400 mg	S, T	-/4 (80)/A	10 (100)	2 (100)	17/18 (94.4)
			Syrup	50 mg/5 mL	S, T	-/3 (60)/NA	8 (80)	2 (100)	13/18 (72.2)
5.	Ceftriaxone	Watch	Injection	250 mg, 500 mg, 1 g	S, T	-/5 (100)/A	10 (100)	2 (100)	18/18 (100)
6.	Azithromycin	Watch	Tablet	250 mg, 500 mg	P, S, T	7 (100)/5 (100)/A	10 (100)	2 (100)	25/25 (100)
			Injection	500 mg	S, T	-/1 (20)/A	4 (40)	2 (100)	8/18 (44.4)
			Suspension	200 mg/5 mL	P, S, T	7 (100)/5 (100)/A	7 (70)	2 (100)	22/25 (88)
7.	Doxycycline	Access	Capsule	100 mg	P, S, T	7 (100)/5 (100)/A	8 (80)	2 (100)	23/25 (92)
8.	Ciprofloxacin	Watch	Tablet	250 mg, 500 mg	P, S, T	7 (100)/5 (100)/A	10 (100)	2 (100)	25/25 (100)
			Injection	200 mg/100 mL	P, S, T	2 (28.6)/1 (20)/A	7 (70)	1 (50)	12/25 (48)
9.	Cotrimoxazole	Access	Tablet	400/80, 800/160 mg	P, S, T	6 (85.7)/4 (80)/A	8 (80)	1 (50)	20/25 (80)
			Suspension	200 + 40 mg/5 mL	P, S, T	1 (14.3)/2 (40)/A	4 (40)	1 (50)	9/25 (36)
10.	Metronidazole	Access	Tablet	200 mg, 400 mg	P, S, T	6 (85.7)/5 (100)/A	10 (100)	2 (100)	24/25 (96)
			Injection	500 mg/100 mL	P, S, T	3 (42.8)/4 (80)/NA	7 (70)	2 (100)	16/25 (64)
			Suspension	200 mg/5 mL	P, S, T	7 (100)/5 (100)/NA	7 (70)	0	19/25 (76)
11.	Vancomycin ^#^	Watch	Injection	250 mg, 500 mg, 1 g	T	-/-/A	6 (60)	1 (50)	8/13 (61.5)
12.	Nitrofurantoin ^#^	Access	Tablet	100 mg	P, S, T	4 (57)/4 (80)/A	8 (80)	2 (100)	19/25 (76)
Antibiotics included in NLEM 2015 but not in the selected (secondary) list for survey									
13.	Ampicillin	Access	Injection	500 mg, 1 g	P, S, T	3 (42.8)/4 (80)/NA	6 (60)	0	14/25 (56)
14.	Benzathine benzylpenicillin	Access	Injection	6 lac, 12 lac Units	P, S, T	0/0/NA	0	0	0
15.	Benzylpenicillin	Access	Injection	10 lac Units	P, S, T	0/0/NA	0	0	0
16.	Cloxacillin ^#^	Access	Capsule	250 mg, 500 mg	P, S, T	2 (28.6)/4 (80)/NA	3 (30)	0	9/25 (36)
			Injection	250 mg	P, S, T	0/1 (20)/NA	1 (10)	0	2/25 (8)
17.	Cefadroxil	Access	Tablet	250 mg, 500 mg	P, S, T	0/2 (40)/NA	7 (70)	0	9/25 (36)
			Syrup	125 mg/5 mL	P, S, T	6 (85.7)/5 (100)/NA	4 (40)	0	15/25 (60)
18.	Cefotaxime	Watch	Injection	250 mg, 500 mg, 1 g	S, T	-/4 (100)/A	7 (70)	2 (100)	14/18 (77.8)
19.	Ceftazidime	Watch	Injection	250 mg, 1 g	S, T	-/1 (20)/NA	6 (60)	0	7/18 (38.9)
20.	Piperacillin + tazobactam ^#^	Watch	Injection	1.125 g, 2.25 g, 4.5 g	T	-/-/A	8 (80)	1 (50)	10/13 (76.9)
21.	Clarithromycin ^#^	Watch	Tablet	250 mg, 500 mg	S, T	-/0/NA	4 (40)	0	4/18 (22.2)
22.	Gentamicin	Access	Injection	10, 40 mg/mL	P, S, T	4 (57)/5 (100)/A	8 (80)	1 (50)	22/25 (88)
Antibiotics included in selected (secondary) list but not in NLEM 2015									
23.	Cefuroxime ^#^	Watch	Injection	750 mg/1.5 g	S, T	-/0/NA	4 (40)	0	4/18 (22.2)
24.	Levofloxacin	Watch	Tablet	500 mg	S, T	-/2 (40)/A	8 (80)	2 (100)	13/18 (72.2)
25.	Norfloxacin	Watch	Tablet	400 mg	P, S, T	4 (57)/3 (60)/A	8 (80)	2 (100)	18/25 (72)
26.	Ofloxacin	Watch	Tablet	100, 200, 400 mg	S, T	-/4 (80)/A	8 (80)	1 (50)	21/25 (84)
27.	Amikacin	Access	Injection	100, 250, 500 mg/2 mL	P, S, T	5 (71.4)/4 (80)/A	10 (100)	2 (100)	22/25 (88)
28.	Linezolid ^#^	Reserve	Tablet	600 mg	T	-/-/A	5 (50)	1 (50)	7/13 (53.8)
			Infusion	2 mg/mL 100, 300 mL	T	-/-/A	4 (40)	0	5/13 (38.4)
29.	Meropenem ^#^	Reserve	Injection	250 mg, 500 mg, 1 g	T	-/-/A	7 (70)	1 (100)	9/13 (69.2)

Data on availability are represented as number (percentage) of facilities where a drug was available on the day of survey. P, S, and T denote the primary, secondary, and tertiary levels of healthcare facilities for which a particular drug is considered “essential”, according to NLEM 2015. * Government-run Affordable Medicines and Reliable Implants for Treatment (AMRIT) pharmacies selling medicines at subsidized costs. ^#^ Antibiotics not included in Haryana state’s essential medicine list (2013–2014). A: available; NA: not available.

**Table 2 antibiotics-13-00131-t002:** Availability of other antimicrobials included in National List of Essential Medicines 2015 and the selected (secondary) list for survey in medicine outlets of different sectors in the district.

S. No.	Name of Antimicrobial	Formulation	Strength	Healthcare Facility Level	Data on Availability			
					Public Sector	Private Sector	Other Sector	Total
					Primary Care (N = 7); Secondary (N = 5); Tertiary (N = 1)	Private Retailers (N = 10)	(AMRIT * Pharmacy; N = 2)	
ANTHELMINTHICS								
1.	Albendazole	Tablet	400 mg	P, S, T	7 (100)/5 (100)/A	10 (100)	2 (100)	25/25 (100)
		Suspension	200 mg/5 mL	P, S, T	7 (100)/4 (80)/A	7 (70)	2 (100)	21/25 (84)
2.	Mebendazole	Tablet	100 mg	P, S, T	0/0/NA	3 (30)	1 (50)	4/25 (16)
3.	Diethylcarbamazine ^#^	Tablet	50 mg, 100 mg	P, S, T	0/0/NA	3 (30)	0	3/25 (12)
4.	Praziquantel ^#^	Tablet	600 mg	S, T	-/0/NA	0	0	0
5.	Ivermectin ^#$^	Tablet	6 mg	P, S, T	3 (42.8)/4 (80)/A	9 (90)	1 (50)	18/25 (76)
ANTIVIRALS								
6.	Acyclovir	Tablet	200 mg, 400 mg	P, S, T	7 (100)/5 (100)/A	10 (100)	2 (100)	25/25 (100)
		Injection	250 mg, 500 mg	S, T	-/5 (100)/A	7 (70)	2 (100)	15/18 (83.3)
		Suspension	400 mg/5 mL	T	-/-/NA	2 (20)	0	2/13 (15.4)
7.	Ganciclovir ^#^	Capsule	250 mg	S, T	-/0/NA	1 (10)	0	1/18 (5.6)
		Injection	500 mg	S, T	-/0/NA	1 (10)	0	1/18 (5.6)
8.	Entecavir ^#^	Tablet	0.5 mg, 1 mg	S, T	-/0/NA	1 (10)	0	1/18 (5.6)
9.	Pegylated IFN α 2a ^#^	Injection	180 µg	S, T	-/0/NA	0	0	0
10.	Pegylated IFN α 2b ^#^	Injection	80, 100, 120 µg	S, T	-/0/NA	0	0	0
11.	Ribavirin ^#^	Capsule	200 mg	S, T	-/4 (80)/A	1 (10)	0	6/18 (33.3)
12.	Sofosbuvir ^#^	Tablet	400 mg	S, T	-/4 (80)/A	1 (10)	0	6/18 (33.3)
13.	Tenofovir ^#^	Tablet	300 mg	S, T	-/3 (60)/A	1 (10)	0	5/18 (27.8)
14.	Oseltamivir ^#$^	Tablet	75 mg	P, S, T	5 (71.4)/4 (80)/A	9 (90)	1 (50)	20/25 (80)
		Syrup	6 mg/mL, 60 mL	P, S, T	5 (71.4)/4 (80)/A	9 (90)	1 (50)	20/25 (80)
ANTIFUNGALS								
15.	Liposomal Amphotericin B	Injection	50 mg/vial	S, T	-/2 (20)/NA	2 (20)	0	4/18 (22.2)
16.	Clotrimazole	Pessary	100 mg, 200 mg	P, S, T	7 (100)/5 (100)/A	8 (80)	2 (100)	23/25 (92)
17.	Fluconazole	Tablet	100, 150, 200, 400 mg	P, S, T	6 (85.7)/5 (100)/A	10 (100)	2 (100)	24/25 (96)
		Injection	2 mg/mL	T	-/-/A	4 (40)	1 (50)	6/13 (46)
18.	Griseofulvin	Tablet	125, 250, 375 mg	P, S, T	1 (14.2)/1 (20)/NA	4 (40)	0	6/25 (24)
19.	Nystatin	Tablet	5 Lac IU	P, S, T	1 (14.2)/0/NA	0	0	1/25 (4)
		Pessary	1 Lac IU	P, S, T	1 (14.2)/0/NA	0	0	1/25 (4)
20.	Itraconazole ^#$^	Tablet	200 mg	P, S, T	6 (85.7)/5 (100)/A	8 (80)	2 (100)	22/25 (88)
21.	Voriconazole ^#$^	Tablet	200 mg	T	-/-/A	6 (60)	1 (50)	8/13 (61.5)
		Injection	200 mg	T	-/-/A	5 (50)	0	6/13 (46)
ANTIPROTOZOALS								
22.	Diloxanide furoate ^#^	Tablet	500 mg	P, S, T	0/0/NA	1 (10)	2 (100)	3/25 (12)
23.	Miltefosine ^#^	Tablet	10, 50 mg	P, S, T	0/0/NA	0	0	0
24.	Paromomycin ^#^	Injection	375 mg/mL	P, S, T	0/0/NA	0	0	0
25.	Artemether + Lumefantrine ^#^	Tablet	20 + 120 mg, 40 + 240 mg, 80 + 480 mg	P, S, T	0/0/NA	6 (60)	0	6/25 (24)
		Dry syrup	20 + 120 mg/5 mL	P, S, T	0/0/NA	1 (10)	0	1/25 (4)
26.	Artesunate ^#^	Injection	60, 120 mg	P, S, T	0/0/NA	7 (70)	1 (50)	8/25 (32)
27.	Artesunate + Sulfadoxine-Pyrimethamine ^#^	Tablet	25 mg + 250/12.5, 50 mg + 55/25	P, S, T	0/0/NA	1 (10)	0	1/25 (4)
28.	Primaquine ^#^	Tablet	2.5, 7.5, 15 mg	P, S, T	1 (14.2)/2 (40)/NA	3 (30)	0	6/25 (24)
29.	Chloroquine ^#^	Tablet	150 mg	P, S, T	3 (42.8)/2 (40)/NA	8 (80)	0	13/25 (52)
30.	Clindamycin ^#^	Capsule	150, 300 mg	P, S, T	1 (14.2)/1 (20)/NA	6 (60)	2 (100)	10/25 (40)
31.	Mefloquine ^#^	Tablet	250 mg	T	-/-/NA	0	0	0
32.	Quinin e ^#^	Tablet	300 mg	P, S, T	1 (14.2)/1 (20)/NA	5 (50)	0	7/25 (28)
		Injection	300 mg/mL	P, S, T	0/0/NA	3 (30)	0	3/25 (12)
33.	Pentamidine ^#^	Injection	200 mg	S, T	-/0/NA	0	0	0
34.	Tinidazole ^$^	Tablet	500 mg	P, S, T	6 (85.7)/5 (100)/A	9 (90)	1 (50)	22/25 (88)

Data on availability are represented as number (percentage) of facilities where a drug was available on the day of survey. P, S, and T denote the primary, secondary, and tertiary levels of healthcare facilities for which a particular drug is considered “essential”, according to NLEM 2015. * Government-run Affordable Medicines and Reliable Implants for Treatment (AMRIT) pharmacies selling medicines at subsidized costs. ^#^ Antimicrobials not included in Haryana state’s essential medicine list (2013–2014). ^$^ Antimicrobials included in the selected list but not in National List of Essential Medicines 2015 (ivermectin, oseltamivir, itraconazole, voriconazole, tinidazole).

**Table 3 antibiotics-13-00131-t003:** Gap analysis for antimicrobial availability according to primary list (NLEM 2015) in public sector.

	Antibiotics	Anthelminthics	Antifungals	Antivirals	Antiprotozoals	Overall Availability in the Facility
PRIMARY CARE						
Facility 1	9 (60)	1 (33.3)	1 (20)	1 (100)	0 (0)	35.3%
Facility 2	10 (66.7)	1 (33.3)	2 (40)	1 (100)	0 (0)	41.2%
Facility 3	10 (66.7)	1 (33.3)	4 (80)	1 (100)	3 (30)	55.9%
Facility 4	7 (46.7)	1 (33.3)	2 (40)	1 (100)	1 (10)	35.3%
Facility 5	6 (40)	1 (33.3)	2 (40)	1 (100)	1 (10)	32.3%
Facility 6	6 (40)	1 (33.3)	2 (40)	1 (100)	1 (10)	32.3%
Facility 7	11 (73.3)	1 (33.3)	2 (40)	1 (100)	1 (10)	47%
*Overall availability in primary care*	56.2%	33.3%	42.8%	100%	10%	
SECONDARY CARE						
Facility 1	13 (65)	1 (25)	2 (40)	3 (42.8)	0 (0)	40.4%
Facility 2	14 (70)	1 (25)	3 (60)	3 (42.8)	3 (27.3)	51%
Facility 3	10 (50)	1 (25)	2 (40)	2 (28.6)	1 (9)	34%
Facility 4	15 (75)	1 (25)	3 (60)	3 (42.8)	1 (9)	48.9%
Facility 5	13 (65)	1 (25)	2 (40)	2 (28.6)	0 (0)	38.3%
*Overall availability in secondary care*	65%	25%	48%	37.1%	9.1%	
TERTIARY CARE						
Facility 1	14 (63.6)	1 (25)	2 (40)	4 (57.1)	0 (0)	42%
*Overall availability*	63.6%	25%	40%	57.1%	0%	
Total overall availability in the district	60.8%	28.9%	44.6%	48.9%	8.7%	

Data are expressed as n (%). A drug was considered “available” at a particular facility if at least one dosage was available. The total number of essential antimicrobials for different healthcare levels as per NLEM 2015 are as follows: antibiotics: 15 (primary care), 20 (secondary care), and 22 (tertiary care); anthelminthics: 3 (primary care) and 4 (secondary and tertiary care); antifungals: 5 (all levels); antivirals: 1 (primary care) and 7 (secondary and tertiary care); and antiprotozoals: 10 (primary care), 11 (secondary care), and 12 (tertiary care).

**Table 4 antibiotics-13-00131-t004:** Gap analysis for antimicrobial availability according to selected (secondary) list in public sector.

Healthcare Level	Antibiotics	Anthelminthics	Antifungals	Antivirals	Antiprotozoals	Overall Availability in the Facility
Primary care						
Facility 1	8 (72.7)	1 (50)	2 (66.7)	2 (100)	1 (100)	72.2%
Facility 2	9 (81.8)	2 (100)	3 (100)	2 (100)	1 (100)	88.9%
Facility 3	11 (100)	2 (100)	3 (100)	2 (100)	1 (100)	100%
Facility 4	9 (81.8)	1 (50)	1 (33.3)	1 (50)	1 (100)	61%
Facility 5	8 (72.7)	1 (50)	2 (66.7)	1 (50)	0	61%
Facility 6	8 (72.7)	1 (50)	2 (66.7)	2 (100)	1 (100)	83.3%
Facility 7	11 (90.9)	2 (100)	3 (100)	2 (100)	1 (100)	100%
*Overall availability in primary care*	83%	71.4%	76.2%	85.7%	85.7%	
Secondary care						
Facility 1	14 (87.5)	2 (100)	3 (75)	2 (100)	1 (100)	87.5%
Facility 2	16 (100)	2 (100)	4 (100)	2 (100)	1 (100)	100%
Facility 3	13 (81)	2 (100)	3 (75)	1 (50)	1 (100)	79.2%
Facility 4	14 (87.5)	1 (25)	4 (100)	2 (100)	1 (100)	87.5%
Facility 5	15 (93.7)	2 (100)	3 (75)	2 (100)	1 (100)	91.7%
*Overall availability in secondary care*	90%	90%	85%	90%	100%	
Tertiary care						
Facility 1	17 (80)	2 (100)	4 (80)	2 (100)	1 (100)	89.6%
*Overall availability in tertiary care*	85%	100%	80%	100%	100%	
*Total overall availability in the district*	86.9%	84%	80.4%	92%	92.3%	

Data are expressed as n (%). A drug was considered “available” at a particular facility if at least one dosage was available. The total number of antimicrobials for different healthcare levels as per the selected (secondary) list for study are as follows: antibiotics: 11 (primary care), 16 (secondary care), and 19 (tertiary care); anthelminthics: 2 (all levels); antifungals: 3 (primary care), 4 (secondary care), and 5 (tertiary care); antivirals: 2 (all levels); and antiprotozoals: 1 (all levels).

## Data Availability

The data presented in this study are available on request from the corresponding author. The data are not publicly available due to privacy concerns.

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
