# Peer review of "The Availability of Essential Antimicrobials in Public and Private Sector Facilities: A Cross-Sectional Survey in a District of North India"

_antibiotics, 2024, doi:10.3390/antibiotics13020131_

Round 1

Reviewer 1 Report

Comments and Suggestions for Authors

 Thi review assess the availability of antimicrobials listed in National List of Essential Medicines in India. I have following concerns while going through this article

1)                   Why the availability of antimicrobials is important?

2)                   What reasons are enlisted in given article for the non availability of anti microbials?

3)                   Which anti microbial drugs were not optimally available in the pharmacies?

4)                   What are the reasons for limited availability of these medications?

5)                   According to WHO define the concept of essential medicines?

6)                   Why the idea of essential medicines is important?

7)                   How we can resolve the issue of lack of regular access to essential medicines?

8)                   How the shortage of pne drug is responsible for anti microbial resistance?

9)                   What are the limitations of this study?

10)               How the given issue can be resolved ?

11)               What is the end result of this whole cross sectional survey?

Comments on the Quality of English Language

some minor changes required

Author Response

Dear Sir/Madam,

All authors thank you for the wonderful and valid comments for improving the manuscript. All authors agree with the comments and have incorporated the comments into the manuscript. Please see below the point wise reply to all the comments.

Reviewer 2 Report

Comments and Suggestions for Authors

Availability and accessibility of essential medicines that satisfy the priority health care needs are necessary. The poor availability of essential antimicrobials could lead to compromised health outcomes and excess economic expenditure in the low-middle-income countries. This study conducted the cross-sectional survey of 25 pharmacies to evaluate the availability of essential antimicrobials and the reasons for their non-availability. The findings of this study provide insight for antibiotic stewardship. But there are still a couple of issues that need to be well addressed:

Introduction

Page 2 Line 47. “In the case of essential antimicrobials, an additional issue of concern with their restricted availability is the emergence of antimicrobial resistance (AMR), a rapidly emerging global threat [5].”

Please clarify the association of restricted availability of essential antimicrobials and the emergence of AMR.

Page 2 Line 69.Data on the availability of antimicrobials from low and middle-income countries (LMICs) 69 is limited and patchy…

Authors referred other survey of essential antimicrobials conducted across multiple LMICs. Please clarify that if there is an overlap of this study with the previous reports, and point out the added value of this study.

Page 2 Line 76.In a country like India where the drug procurement system in the public sector is quite different from Western countries and the supply of medicines is mainly governed by demand and prescription patterns”.

Please provide a further description of the difference of drug procurement system that may provide readers a better context for understanding the study.

Page 2 Line 79. “With this background, the present study was planned to generate information on the availability of essential anti-microbials listed in NLEM 2015 in public as well as private

sectors in a district of North India and to identify the reasons for their non-availability.”

Please revise “anti-microbials” into “antimicrobials”.

Results

Page 3 Line 83.A total of 25 pharmacies comprising 13 public (7 primary, 5 secondary, and 1 tertiary healthcare), 10 private, and 2 other sector pharmacies were surveyed in the district.

The description of distribution of sample pharmacies in rural and urban would be helpful to understand the representativeness of sampling.

Table 1 and 2.

a. Please added the explanation for “7 (100)/ 5 (100)/ A” in the table.

b. Please added the explanation for the abbreviations “P,S,T” in the footnote of table.

c. The decimal precision in the tables should be consistent.

Page 10 Line 83.Most of the pediatric formulations of antibiotics including amoxicillin…

Please provide the definition of “pediatric formulations of antibiotic”. Did authors refer “pediatric formulations” to Suspension and Syrup Formulation?

Table 3.

The decimal precision in the tables should be consistent.

Figure 1

The vertical axis scale should range from 0 to 100.

Discussion

Page14 Line 98.Of note, there was optimal availability of most access group of antibiotics across public and private sectors.

This statement might confuse readers since you stated that the availability of access group was limited in the Page14 Line 95: “Limited availability of access groups and optimal availability of watch groups are two sides of the same coin…

Page15 Line 136.…non-listing of these drugs in Haryana state EML being the quoted reasons for non-availability.

“Not a part of state EML” is one of important reason for non-availability of antimicrobials in public and private sector. Further discussions on the discrepancy of NEML with Haryana state EML could strengthen the explanations of results.

Materials and Methods

Page17 Line 194. The statement and proportion you mentioned are unclear: “Nearly 70 percent expenditure on medicines is borne by patients out-of-pocket; reported as 59 to 86 percent in public sector in the three North Indian states of Haryana, Punjab and Chandigarh.

Page 17 Line 225. “The facilities to be surveyed were selected by systematic random sampling. For this, the sampling interval was calculated as the total number of pharmacies (in rural, urban and semi-urban areas in the district) in the sampling frame (N) divided by the sample of interest (n=5 for primary and secondary care public pharmacies; 10 for private pharmacies, and 5 for another sector).”

Please provide the total number of pharmacies in your sample area. And please explain why you set the number 5, 10, 5 for your survey. It is unclear if the sampling intervals and sample size was determined based on a power calculation, which could affect the precision and generalizability of the findings. What is the representativeness of your survey sample? A more detailed explanation of the sampling intervals calculation would be helpful.

Page18 Line 243.The data on availability of antimicrobials was collected with respect to antimicrobials listed as essential in NLEM 2015 (primary list) …

I agree with using NLEM 2015 as primary list to evaluate the availability of antimicrobials. But it would provide the latest insights if author provided the overlap of primary list with NLEM 2022 in supplements.

Page18 Line 245.selected (secondary) list of antimicrobials which was prepared by consensus among study investigators after reviewing the agents indicated for commonly encountered infectious illnesses in various healthcare settings (by discussion with medical officers and physicians).

Please provide more details on the selection of the “secondary list of antimicrobials”.

Comments on the Quality of English Language

Minor editing of English language required

Author Response

(The authors gave the same response as above.)

Reviewer 3 Report

Comments and Suggestions for Authors

1-  This is an excellent review on a topic that is of much importance, basically, the availability of antimicrobials in a  district in North India.

2-  This is a recurring theme in various districts and regions around the world.

3-  The author presented a clear and comprehensive documentation of the availability of antibiotics, antimicrobials, antihelminthics, etc., as seen in tables 1-3. 

4- Also of importance are Figure 1 (Gap analysis for Availability of antibiotics in public sectors according of WHO, as well as Reasons for Nonavailability of antimicrobials in public and private sectors. Both of these observations are important.

5- The references are up-to date, and well documented.

Author Response

Dear Sir/Madam,

All authors thank you for the wonderful and valid comments.  

Round 2

Reviewer 1 Report

Comments and Suggestions for Authors

I am satisfied with authors reply